# Offspring production from cryopreserved primordial germ cells in *Drosophila*

Miho Asaoka [1,5]✉, Yurina Sakamaki[2,5], Tatsuya Fukumoto[3,5], Kaori Nishimura[4], Masatoshi Tomaru [4], Toshiyuki Takano-Shimizu [4]✉, Daisuke Tanaka [3]✉ & Satoru Kobayashi [1,2]✉

There is an urgent need to cryopreserve *Drosophila* stocks that have been maintained as living cultures for a long time. Long-term culture increases the risk of accidental loss and of unwanted genetic alteration. Here, we report that cryopreserved primordial germ cells (PGCs) can produce F1 progeny when transplanted into hosts. The cryopreserved donor PGCs could form germline stem cells in host gonads and contributed to continuous offspring production. Furthermore, the ability to produce offspring did not appear to vary with either differences between donor strains or cryopreservation duration. Therefore, we propose that our cryopreservation method is feasible for long-term storage of various *Drosophila* strains. These results underscore the potential usefulness of our cryopreservation method for backing up living stocks to avoid either accidental loss or genetic alteration.

[1] Life Science Center for Survival Dynamics, Tsukuba Advanced Research Alliance (TARA), University of Tsukuba, Tsukuba, Ibaraki 305-8577, Japan.
[2] Graduate School of Life and Environmental Sciences, University of Tsukuba, Tsukuba, Ibaraki 305-8572, Japan. [3] Research Center of Genetic Resources, National Agriculture and Food Research Organization (NARO), Tsukuba, Ibaraki 305-8602, Japan. [4] Advanced Insect Research Promotion Center, Kyoto Institute of Technology, Kyoto 616-8354, Japan. [5]These authors contributed equally: Miho Asaoka, Yurina Sakamaki, Tatsuya Fukumoto.
✉email: masaoka@tara.tsukuba.ac.jp; fruitfly@kit.ac.jp; dtanaka@affrc.go.jp; skob@tara.tsukuba.ac.jp

Model organisms that carry chromosomal aberrations, mutations, and transgenic constructs are critical for studies of gene functions in animal systems. The fruit fly, *Drosophila melanogaster*, is a classical model animal for basic biology research. Use of the fly has enabled key advances in our understanding of human diseases as well as a wide range of phenomena in developmental, cellular, and evolutionary biology[1–3]. The importance of this animal to biological research is due largely to its ease of culture and genetic manipulation[4,5]. The number of *Drosophila* strains harboring mutations and transgenes continues to increase, and strains have been maintained as living cultures in both individual laboratories and *Drosophila* stock centers. However, long-term maintenance of *Drosophila* strains as living stocks increases the risk of unwanted second mutations, which can cause phenotypes to be altered or important strains to be lost. Cryopreservation is widely used to preserve genetic strains of model organisms, although not for flies[6–12]. Hence, cryopreservation techniques are urgently needed for long-term storage of *Drosophila* strains to prevent accidental loss of genetic strains due to labor shortages induced by the pandemic and decrease the risk of genetic alteration.

*Drosophila* strains can be preserved by freezing ovaries or embryos[13–15]. However, these methods are laborious, have variable success rates, and have poor reproducibility. Consequently, these methods are not practical for preserving a variety of *Drosophila* strains[16]. In order to overcome this critical limitation, we sought to cryopreserve primordial germ cells (PGCs) instead of ovaries and embryos. When transplanted into host animals, PGCs can differentiate into gametes that are subsequently fertilized to produce offspring[17,18].

## Results and discussion

PGCs were collected from donor embryos at the blastoderm stage (stage 5) using a thin glass needle and subsequently suspended in cryoprotectant agent (CPA) for cryopreservation in liquid nitrogen ($LN_2$) (see Methods). We first optimized the chemical composition of the CPA. PGCs were immersed in CPAs that all contained either ethylene glycol (EG), dimethyl sulfoxide (DMSO), or glycerol (G), along with sucrose at various concentrations. We examined their morphology immediately after freeze-thawing (Supplementary Table 1 and Supplementary Fig. 1). Sucrose was used to reduce the risk of intracellular ice crystal formation due to dehydration[19]. Without using CPA, all PGCs were ruptured and indiscernible after freeze-thawing (Supplementary Table 1 and Supplementary Fig. 1). However, more than 50% of PGCs remained discernible when suspended in CPA containing 20% EG and 1 M sucrose (Supplementary Table 1 and Supplementary Fig. 1). We therefore used this composition of CPA throughout this study.

During normal development, PGCs migrate through embryos to reach the embryonic gonads, where they differentiate into gametes at the post-embryonic stage. Hence, we next examined whether PGCs that had been CPA-treated and freeze-thawed (F-PGCs) could enter embryonic gonads. When F-PGCs obtained from donor embryos that express green fluorescent protein (GFP) in the germline throughout development (*EGFP-vas* embryos)[20] were transplanted into host embryos with *yellow* and *white* mutations (*y w* host embryos), GFP-labeled PGCs were detected in the gonads of 64.3% of host embryos (Fig. 1a and Supplementary Table 2). This result indicates that F-PGCs retain the ability to migrate into embryonic gonads.

By contrast, PGCs that were treated with CPA but not freeze-thawed (CPA-PGCs) were able to colonize the gonads of all *y w* host embryos (Supplementary Table 2). Similar results were obtained with PGCs that had not been subjected to CPA-treatment and freeze-thawing (Naive-PGCs) (Supplementary Table 2). These data indicate that freeze-thawing, but not CPA-treatment, reduces the frequency of embryos with donor PGCs within embryonic gonads. In embryonic gonads carrying donor PGCs, however, similar numbers of donor PGCs were observed regardless of freeze-thawing (Fig. 1b). This similarity may be due to severe reduction in the viability of F-PGCs sometimes caused by freeze-thawing following transplantation into *y w* host embryos. Thus, we propose that almost all F-PGCs that are improperly freeze-thawed are eliminated in the host, whereas F-PGCs that are properly treated retain a similar ability to survive and migrate into the gonads, comparable to CPA-PGCs and Naive-PGCs.

We next asked whether F-PGCs can produce the next generation of progeny. PGCs marked by an *EGFP-vas* transgene containing the *white* ($w^+$) gene were transplanted into *y w* host embryos. Host embryos were allowed to develop to adulthood and the resultant adults were mated with $w^-$ flies. F1 progeny derived from donor PGCs were expected to have red eyes ($w^+$ phenotype), whereas the host germline produces white-eyed offspring ($w^-$ phenotype). We found that 35.5% of adult hosts transplanted with F-PGCs produced red-eyed F1 progeny (Table 1), but untransplanted hosts did not. This result indicates that the F-PGCs retained the ability to produce offspring. However, a lower percentage of hosts produced donor-derived progeny when transplanted with F-PGCs compared with hosts transplanted with either CPA-PGCs or Naive-PGCs (Table 1). This is compatible with our observation that F-PGCs colonized embryonic gonads less frequently than CPA-PGCs or Naive-PGCs.

Supplementary Table 3 shows that F-PGCs can produce functional gametes of both sexes. Since heterosexual transplantation of PGCs results in failure to produce gametes[21,22], we usually transplanted PGCs collected from 6–35 embryos (27 on average) into host embryos (see Methods). This procedure enabled us to transplant a mixture of female and male PGCs into a host, without sexing of donor and host embryos. We found that F-PGCs transplanted into male *y w* hosts produced F1 progeny of both sexes (Supplementary Table 3). This ability is particularly important considering that male-specific Y-chromosomes of donors can be obtained only from male F1 progeny derived from male F-PGCs.

We next determined whether F-PGCs can give rise to germline stem cells (GSCs) in the adult gonads for continuous production of F1 progeny. We identified donor-derived GSCs as single cells that express GFP, associated with niche cells (cap cells in ovaries, and hub cells in testes), and were functional. GSC function for continuous production of differentiating cysts was assessed using a lineage tracer, GFP; GFP-positive single cells were followed by a series of differentiating cysts that also produced the lineage tracer (Fig. 1c). We found that F-PGCs can become GSCs in adult gonads when transplanted into *y w* hosts (Fig. 1c). The number of GSCs derived from F-PGCs was similar to GSCs originating from CPA-PGCs or Naive-PGCs in both females and males (Fig. 1d).

The presence of functional GSCs derived from F-PGCs in females was further supported by the following observation. All females carrying GSCs differentiated from F-PGCs continued to produce donor-derived F1 progeny for at least 4–6 days after mating (Fig. 1e, Supplementary Fig. 2, and Supplementary Table 4). Moreover, the number of donor-derived F1 progeny produced by females carrying donor-derived GSCs was similar among the females transplanted with F-PGCs, CPA-PGC, and Naive-PGCs (Fig. 1f). Thus, we speculate that F-PGCs, CPA-PGCs, and Naive-PGCs are almost equally capable of becoming GSCs once they colonize embryonic gonads.

To determine the feasibility of our cryopreservation method, we went on to ask if long-term storage of PGCs in $LN_2$ affects their

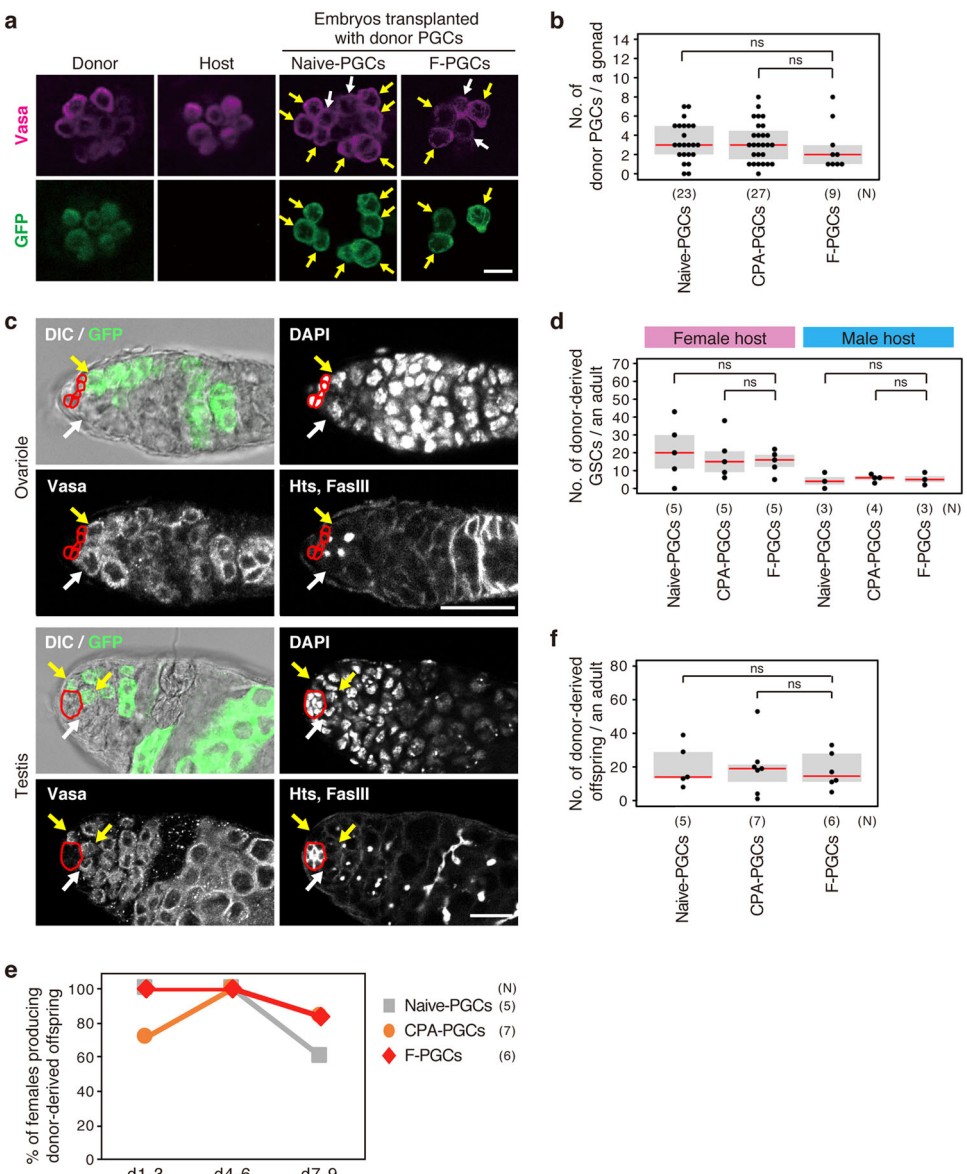

**Fig. 1 F-PGCs can migrate into the gonads and give rise to GSCs. a** Representative images of PGCs in the gonads of *EGFP-vas* donor embryos (Donor), *y w* host embryos (Host), and *y w* embryos transplanted with PGCs without both CPA-treatment and freeze-thawing (Naive-PGCs) or treated with CPA and freeze-thawed (F-PGCs). Stage 15 embryos were double-stained for a germline-marker, Vasa (magenta), and for GFP (green). Yellow and white arrows indicate GFP-positive donor PGCs and GFP-negative host PGCs in *y w* host embryos, respectively. **b** Donor-derived PGCs in the gonads of *y w* host embryos transplanted with Naive-PGCs, PGCs treated with CPA but not subject to freeze-thawing (CPA-PGCs), and F-PGCs were counted. Each dot represents the number of GFP-positive PGCs per gonad. **c** Representative images of ovariole and testis with F-PGC–derived germline. Ovaries and testes were dissected from adult *y w* hosts 10 days after eclosion (Supplementary Fig. 2), and stained for GFP (green), Vasa, Hts (a spectrosome/fusome marker), FasIII, and nuclei (DAPI). FasIII stains pre-follicle cells in ovaries, and hub cells in testes. DIC image merged with GFP signal is shown (top left). Images for Vasa (bottom left), DAPI (top right), and Hts and FasIII signals (bottom right) are also shown. Yellow and white arrows indicate F-PGC–derived GSCs and host GSCs, respectively. GSC niche cells, cap cells in the ovary, and hub cells in the testis are outlined in red. **d** Donor-derived GSCs were counted in adult *y w* hosts producing F1 progeny derived from Naive-PGCs, CPA-PGCs, or F-PGCs. Each dot represents the number of GFP-positive GSCs per adult host. See Supplementary Fig. 2. for details. **e** Percentage of female *y w* hosts carrying GSCs derived from Naive-PGCs (gray), CPA-PGCs (orange), or F-PGCs (red) producing donor-derived F1 progeny on days 1–3 (d1–3), 4–6 (d4–6), and 7–9 (d7–9) after mating. **f** Donor-derived F1 progeny produced from *y w* female host carrying GSCs derived from Naive-PGCs, CPA-PGCs, and F-PGCs were counted. Each dot represents the number of donor-derived progeny produced from each female host on days 1–9 after mating. "N" represents the number of gonads (**b**) and adults (**d**–**f**) examined. "ns" indicates not significant ($P > 0.1$, Wilcoxon test) in **b**, **d**, and **f**. In **b**, **d**, and **f**, red bars represent median values. The upper and lower borders of the box show the 75% and 25% quartiles, respectively. Scale bars, 10 μm (**a**) and 20 μm (**c**).

ability to produce F1 progeny. When PGCs obtained from *EGFP-vas* donors were stored in LN$_2$ for 8–30 and 31–150 days (*EGFP-vas* 8–30d and 31–150d in Fig. 2a, b), the percentage of adult *y w* hosts producing donor-derived F1 progeny, and the offspring number were statistically similar to that obtained with F-PGCs

(*EGFP-vas* 0d in Fig. 2a, b). These results indicate that PGCs retain their ability to produce F1 progeny even after being cryopreserved for 31–150 days.

We next determined whether this cryopreservation method can be used on various genetic strains. Donor PGCs obtained from

**Table 1 Production of offspring derived from donor PGCs.**

| Donor PGCs[a] | No. of embryos transplanted | No. of adults eclosed | No. of adults producing donor-derived F1[b] [Females, Males] (%) | | PGC transplantation efficiency[c] |
|---|---|---|---|---|---|
| None | — | 89 | 0 | [0, 0] (0.0)* | |
| F-PGCs | 125 | 31 | 11 | [6, 5] (35.5) | 8.8% |
| CPA-PGCs | 67 | 19 | 13 | [8, 5] (68.4)* | 19.4% |
| Naive-PGCs | 141 | 29 | 19 | [10, 9] (65.5)* | 13.5% |

[a]Donor PGCs obtained from EGFP-vas embryos were CPA-treated and freeze-thawed (F-PGCs), treated with CPA but not freeze-thawed (CPA-PGCs), or free from CPA-treatment and freeze-thawing (Naive-PGCs). These donor PGCs were transplanted into y w host embryos, and allowed to develop to adulthood.
[b]The numbers of female and male adults producing donor-derived F1 progeny were counted. The percentage of adults producing donor-derived offspring is shown in parentheses. Significance was calculated vs. F-PGCs (*$P < 0.05$) using a two-sided Fisher's exact test.
[c]PGC transplantation efficiency measured by the number of single adult hosts producing donor-derived F1 progeny (no. of adults producing donor-derived F1/no. of transplanted embryos).

three attP strains (attP-3B_00033, attP40, and attP-3B_00037) were stored in $LN_2$ for 31–150 days. The percentages of adult y w hosts producing donor-derived F1 progeny and offspring numbers were similar to hosts with donor PGCs that were taken from EGFP-vas embryos and stored in $LN_2$ for 31–150 days (Fig. 2a, b). Thus, the ability to produce offspring does not appear to vary with donor strain. Furthermore, even when donor PGCs obtained from wild-type embryos (BER_2 strain) were maintained in $LN_2$ for a longer duration (360–400d) they were likewise capable of contributing to offspring production (Fig. 2a, b). These data strongly suggest that our cryopreservation method is feasible for long-term storage of PGCs from various fly strains.

Success rates for cryopreservation of PGCs did not vary among either lab workers or laboratories. The cryopreservation protocol was originally developed by a group at University of Tsukuba (UT), and was subsequently transferred to Kyoto Institute of Technology (KIT). The percentage of y w hosts producing donor-derived F1 progeny and offspring numbers obtained by the UT group (EGFP-vas_0d) were statistically similar to those obtained by the KIT group (EGFP-vas 8–30d, 31–150d, attP-3B_00033 31–150d, attP40 31–150d, attP-3B_00037 31–150d, and BER_2 360–400d), regardless of donor strains and duration of PGC maintenance in $LN_2$ (Fig. 2a, b); however, the offspring numbers obtained by the UT group were larger than those obtained by the KIT group (Fig. 2b). The learning period for our cryopreservation technique depends on the individual lab worker's experiences, ranging from one week to three months (Supplementary Table 5).

Because strains carrying multiple mutations or chromosomal aberrations such as deficiencies, translocations, duplications, and insertions are difficult to reproduce with the CRISPR/Cas9 technique[5], and also because weak or unhealthy stocks are difficult to maintain as living cultures, cryopreservation of such strains is urgently needed. When y w embryos are used as hosts to obtain progeny from cryopreserved PGCs, it is necessary to distinguish the donor-derived progeny from host-derived progeny using either dominant genetic markers or PCR. Furthermore, it is extremely difficult to reconstitute the original chromosome constitution from donor-derived progeny by mating. In order to reduce the amount of labor required for the selection and mating steps, we therefore used an agametic host.

We have previously reported that embryos over-expressing ovo-A mRNA in PGCs using germline-specific nanos-Gal4 driver (OvoA_OE embryos) fail to produce gametes in both sexes[23]. We found that OvoA_OE embryos became sterile unless donor PGCs were transplanted (Table 2). When F-PGCs were transplanted into OvoA_OE host embryos from donor embryos of a weak strain (M17) harboring five mutations and a balancer chromosome (M17 0d PGCs), 44.4% of female hosts and 44.4% of male ones were fertile (Table 2). F1 progeny produced by inbreeding among these fertile hosts, and F2 progeny produced by mating

among the F1 progeny showed the phenotype identical to that of M17 (Table 3). Furthermore, the number of F2 flies originated from M17 0d PGCs was almost identical to the progeny number obtained by the original M17 strain (Fig. 2d). Thus, the agametic host enabled us to revive the original donor strains from the donor F-PGCs by inbreeding.

Next, we determined whether long-term cryopreservation of PGCs affects their ability to produce progeny in agametic host, when transplanted into OvoA_OE hosts. When PGCs obtained from M17 embryos were cryopreserved for 8–30 and 31–150 days and transplanted, the fraction of fertile hosts and the number of F1 and F2 progeny were similar to those obtained with M17 0d PGCs (Table 2 and Fig. 2c, d). Furthermore, all F1 and F2 progeny derived from the long-term cryopreserved PGCs were phenotypically identical to the donor M17 strain (Table 3). These results indicate that our cryopreservation method using an agametic host can revive the original strain, even after long-term storage.

Our results indicate that F-PGCs retain the ability to migrate into embryonic gonads. Once they enter the host gonads, they normally give rise to GSCs and produce donor-derived F1 progeny of both sexes, thereby enabling us to restore the donor strain. These abilities of F-PGCs are retained when they are transplanted into agametic host embryos. Table 2 shows that, at most, 15 transplanted host embryos were required to obtain single host couple producing donor-derived progeny of both sexes [no. of embryos transplanted/no. of couples producing donor-derived F1 progeny: 74/5 ~ 15 (M17 31–150d)]. Given that 10–20 PGCs were transplanted into a host embryo (see Method), 150–300 PGCs are required to produce a single fertile couple, and this number of PGCs can be collected into one glass needle from 27 donor embryos. This PGC collection and subsequent PGC transplantation into agametic host embryos can be performed within 50 and 100 min (min), respectively.

Our data underscore the potential usefulness of our cryopreservation method for backing up living stocks, as evidenced by the fact that the KYOTO Stock Center at KIT has started to cryopreserve PGCs from some Drosophila strains that had previously been maintained by living culture. To propagate our technique, we are preparing a video-based report on our cryopreservation protocol using the agametic host. At present, Drosophila researchers can choose between at least two known cryopreservation methods for fly stocks. Zhan et al. recently reported a new method for cryopreserving Drosophila embryos[24]. However, according to their method, only embryos that fall within a narrow range of ages optimal for cryopreservation should be used[24], which can be challenging to collect. By contrast, our method uses PGCs obtained from blastodermal embryos, which can be easily collected. Thus, our PGC-based cryopreservation method may be more advantageous than cryopreserving embryos. In the future, we will compare the efficiency and feasibility of these two cryopreservation methods.

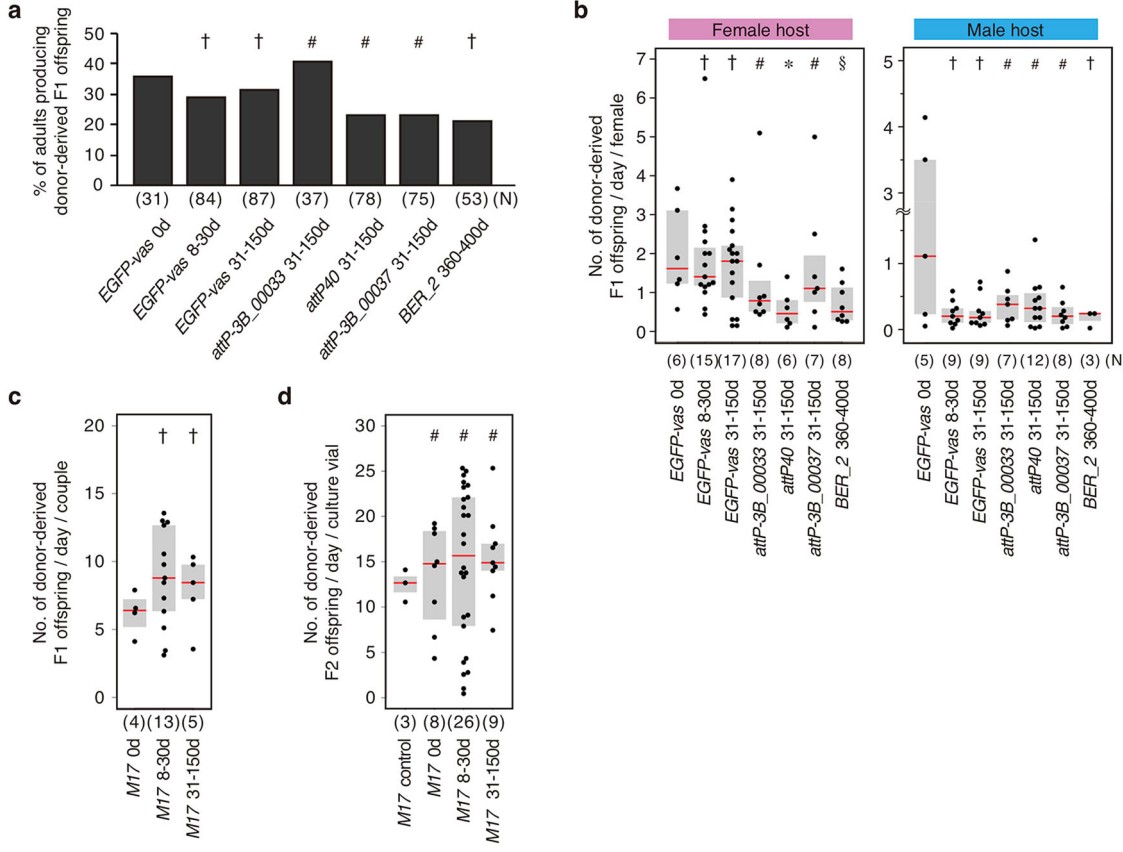

**Fig. 2 Long-term cryopreservation of PGCs from various strains. a** Percentage of adult $y\ w$ hosts producing donor-derived offspring. Each adult female or male host was mated with 5 $y\ w$ males or 4–5 $y\ w$ females, respectively. Females were allowed to lay eggs for 9 days (on days 1–9 after mating), and adult hosts producing donor-derived F1 progeny with red eyes were scored. The adult $y\ w$ hosts developed from embryos transplanted with F-PGCs (*EGFP-vas* 0d) or PGCs cryopreserved in LN₂ for 8–30 days (*EGFP-vas* 8–30d) and 31–150 days (*EGFP-vas* 31–150d) obtained from *EGFP-vas* donor embryos. PGCs cryopreserved for 31–150 days that were obtained from donor embryos from *attp-3B VK00033* (*attp-3B_00033* 31–150d), *attp_40* (*attp_40* 31–150d), *attp-3B VK00037* strain (*attp-3B_00037* 31–150d), and wild-type PGCs (*BER_2*) cryopreserved for 360–400 d (*BER_2* 360–400d) were also tested. "N" shows the number of adults examined. Significance was calculated vs. *EGFP-vas* 0d (†$P > 0.05$), and vs. *EGFP-vas* 31–150d (#$P > 0.05$) using a two-sided Fisher's exact test. **b** Donor-derived F1 progeny produced from the above female and male $y\ w$ hosts were counted (see Methods). Each dot represents the number of F1 progeny produced from each female per day produced from each adult host. "N" is the number of adults examined. Significance was calculated vs. *EGFP-vas* 0d (†$P > 0.05$, §$0.02 < P < 0.05$), and vs. *EGFP-vas* 31–150d (#$P > 0.05$, *$0.02 < P < 0.05$) using the Wilcoxon test. **c** Donor (*M17*)-derived F1 progeny produced from OvoA_OE hosts transplanted with F-PGCs (*M17* 0d), or *M17* PGCs cryopreserved for 8–30 days (*M17* 8–30d) and 31–150 days (*M17* 31–150d) were counted. Each dot represents the number of F1 progeny produced from each OvoA_OE host couple per day. "N" is the number of couples examined. Significance was calculated vs. *M17* 0d (†$P > 0.05$) using the Wilcoxon test. See Methods for details on progeny production. **d** F1 progeny derived from OvoA_OE hosts transplanted with F-PGCs (*M17* 0d), or *M17* PGCs cryopreserved for 8–30 days (*M17* 8–30d) and 31–150 d (*M17* 31–150d) were inbred to produce F2 progeny. Eight to ten F1 females and 5–10 F1 males were cultured in a vial, and were allowed to lay eggs for 9 days. Each dot represents the number of F2 progeny eclosed in each vial per day. As a control, 10 females and 10 males from *M17* strain were mated (*M17* control). "N" shows the number of vials examined. Significance was calculated vs. *M17* control (#$P > 0.05$) using the Wilcoxon test. In **b**, **c**, and **d**, red bars represent median values. The upper and lower borders of the box indicate the 75% and 25% quartiles, respectively.

## Methods

**Drosophila stocks**. $y\ w$ was used as a source of host embryos for PGC transplantation. We used OvoA_OE embryos derived from *nanos-Gal4VP16* (*nanos-Gal4*) females (a gift from R. Lehmann)[25] mated with *UASp-Ovo-A* (line#4-2) males[23] as agametic hosts. The following strains were used as sources of donor embryos: $w^*$; $P\{vas.EGFP.HA\}$ (KYOTO Stock Center, Stock No. 109171)[20], referred to as *EGFP-vas*, $y1\ M\{vas\text{-}int.Dm\}ZH\text{-}2A\ w^*$; $PBac\{y^+\text{-}attP\text{-}3B\}VK00033$ (KYOTO Stock Center, Stock No. 130448)[26], designated as *attP-3B_00033*, $y^1\ M\{vas\text{-}int.Dm\}ZH\text{-}2A\ w^*$; $PBac\{y^+\text{-}attP\text{-}3B\}VK00037$ (KYOTO Stock Center, Stock No. 130449)[26], designated as *attP-3B_00037*, $y^1\ v^1\ P\{y^{+t7.7} = nos\text{-}phi\text{-}C31\backslash int.NLS\}X$; $P\{y^{+t7.7} = CaryP\}attP40$ (BDSC, Stock No. 25709), designated as *attP40*, wild-type strain *BER_2* (BDSC, Stock No.3840), and $y\ w$; $Pri[1]/TM6B$, $P\{Dfd\text{-}GMR\text{-}nvYFP\}4$, $Sb[1]\ Tb[1]\ ca[1]$, designated as *M17*. All stocks were maintained at 25 °C on standard *Drosophila* culture medium.

**Preparation of glass needles**. Needles were made from glass capillaries (25-μl Drummond Microcaps) using a needle puller (Narishige), and the tip was sharpened using a needle grinder (Narishige). The needle gauge was adjusted to 12–14

μm. A micromanipulator (Leica) equipped with a glass needle was used for transplantation.

**Freeze-thawing of PGCs**. Donor and host embryos were collected at 50-min intervals, and were allowed to develop until 100–150 min after egg laying (AEL). Collected embryos were dechorionated, and aligned on double-sided sticky tape as previously described[18]. In silicone oil (FL-100-450CS, Shin-Etsu Chemical), 100–470 PGCs (178 on average) were collected into one glass needle from 6–35 donor embryos (27 on average). PGCs were obtained from donor embryos at early stage 5 (130–150 min AEL), when PGC formation has finished, but not completed, cellularization of soma[27]. We could therefore obtain PGCs while avoiding contamination by somatic cells. The PGCs were later suspended in an equal volume of CPA [Ephrussi–Beadle Ringer solution (EBR)[28] containing 20% ethylene glycol, and 1 M sucrose]. The glass needle containing the suspended PGCs was dipped in liquid nitrogen (LN₂) for 20 seconds (sec). For long-term cryopreservation, the glass needle containing the PGCs was dipped, and maintained in LN₂ for either 8–30, 31–150, or 360–400 days. To thaw the PGCs, the needle was dipped in silicone oil at 25 °C for 10 sec. Next, needle contents were placed in silicone oil

**Table 2 Production of offspring derived from donor PGCs by agametic hosts.**

| Donor PGCs[a] | No. of embryos transplanted | No. of fertile adults[b]/eclosed adults (%)[c] | | | | No. of couples producing F1[d] | PGC transplantation efficiency[e] |
|---|---|---|---|---|---|---|---|
| | | Females | | Males | | | |
| None | — | 0/43 | (0.0)* | 0/46 | (0.0)* | 0 | |
| M17 0d | 33 | 4/9 | (44.4) | 4/9 | (44.4) | 4 | 12.1% |
| M17 8–30d | 134 | 13/27 | (48.1)$^{ns}$ | 7/30 | (23.3)$^{ns}$ | 13 | 9.7% |
| M17 31–150d | 74 | 5/20 | (25.0)$^{ns}$ | 3/10 | (30.0)$^{ns}$ | 5 | 6.8% |

[a]OvoA_OE hosts were transplanted with F-PGCs (M17 0d) or M17 PGCs cryopreserved for 8–30 days (M17 8–30d) and 31–150 days (M17 31–150d) that were obtained from M17 donor embryos. After transplantation, OvoA_OE hosts were allowed to develop to adulthood.
[b]Fertility of female and male hosts was determined as described in Methods.
[c]The percentage of adult hosts producing offspring is shown in parentheses. Significance was calculated vs. M17 0d (*P < 0.05, ns: P > 0.1) using a two-sided Fisher's exact test.
[d]The number of couples is shown (Supplementary Fig. 3).
[e]The transplantation efficiency (no. of host couples producing F1 progeny/no. of transplanted embryos). These values were not significantly different, compared with M17 0d (P > 0.1, two-sided Fisher's exact test).

**Table 3 Genotype of F1 and F2 progeny produced by host couples.**

| Donor PGCs[a] | No. of F1 progeny examined | | | No. of F2 progeny examined | | |
|---|---|---|---|---|---|---|
| | Total[b] [Females, Males] | | With identical phenotype to donor[c] (%) | Total[d] [Females, Males] | | With identical phenotype to donor[c] (%) |
| M17 0d | 223 | [122, 101] | 223 (100.0) | 964 | [503, 461] | 964 (100.0) |
| M17 8–30d | 1035 | [501, 534] | 1035 (100.0) | 3435 | [1791, 1644] | 3435 (100.0) |
| M17 31–150d | 354 | [168, 186] | 354 (100.0) | 1258 | [666, 592] | 1258 (100.0) |

[a]OvoA_OE hosts were transplanted with F-PGCs (M17 0d) or M17 PGCs cryopreserved for 8–30 days (M17 8–30d) and 31–150 days (M17 31–150d) that were obtained from M17 donor embryos. After transplantation, OvoA_OE hosts were allowed to develop to adulthood.
[b]The number of F1 progeny produced from the couples shown in Table 2. The couples were allowed to lay eggs for 9 days. See Supplementary Fig. 3 for details.
[c]The number of F1 and F2 progeny showing M17 phenotype is shown. The adult flies from M17 strain show yellow (body color), white (eye color), Prickly (bristle morphology), Stubble (bristle morphology), and Tubby (body shape) phenotypes. Prickly and Stubble phenotypes act as markers for the presence of balancer chromosome, TM6. We also examined 180 females and 156 males (336 total) flies from M17 strain, and found that all of the adult flies show these mutant phenotypes.
[d]Total number of F2 progeny examined is shown. Eight to ten F1 females and 5–10 F1 males were cultured in a vial and were allowed to lay eggs for 9 days. The eggs were developed to adults, and the number of F2 progeny was counted. See Supplementary Fig. 3 for details.

under a compound microscope, and PGCs were collected again using a needle with a minimal amount of CPA. PGCs within the needle were transplanted into either 2–30 y w or OvoA_OE host embryos (10 on average).

**PGC transplantation.** PGC transplantation was conducted at 25 °C as previously described[18,29]. Freeze-thawed PGCs were transplanted into the posterior pole of either y w host embryos or OvoA_OE host embryos at the cellular blastoderm stage (100–150 min AEL). Between 10 and 20 PGCs were injected into each host embryo. The number of PGCs transplanted into host embryos was counted under a microscope. Transplanted embryos were maintained at 25 °C until hatching. Hatched larvae were transferred into a standard culture medium in 35 mm culture dishes (Falcon brand, Corning Life Sciences, Schiphol-Rijk, Netherlands), and incubated at 25 °C until pupation (<15 larvae per dish). The resultant pupae were transferred into vials containing culture medium and maintained at 25 °C until eclosion.

**Effect of CPA on PGC morphology.** PGCs obtained from EGFP-vas embryos were suspended in an equal volume of CPA (EBR containing ethylene glycol, dimethyl sulfoxide, or glycerol along with sucrose at various concentrations), and freeze-thawed as described above. PGCs were subsequently placed on the surface of embryos in silicone oil. We counted the PGCs before and after freeze-thaw treatment and compared their numbers.

**Immunostaining.** For double-staining of embryos for GFP and Vasa, transplanted embryos were left to develop at 18 °C for 20 h. Under a dissection microscope, the developmental stages of the transplanted embryos were determined based on morphology[27]. Stage 15 embryos were transferred into a tube. Silicone oil was removed by washing the embryos with heptane. Washed embryos were subsequently fixed and double-stained with rabbit anti–GFP antibody (diluted 1:500, Molecular Probes) and chicken anti–Vasa antibody (1:500, lab stock), as previously described[30,31]. Primary antibodies were detected using Alexa Fluor 488–conjugated anti–rabbit IgG antibody A-11034 (1:500, Molecular Probes) and Cy3–conjugated anti–chick IgY antibody (1:500, Jackson ImmunoResearch).

Antibody staining of ovaries and testes was performed as previously described[31]. Rabbit anti–GFP antibody (1:500, Molecular Probes), chick anti–Vasa antibody (1:500, lab stock), mouse anti–Hts antibody 1B1 [1:10, Developmental Studies Hybridoma Bank (DSHB)], and mouse anti–FasIII antibody 7G10 (1:10,

DSHB) were used as primary antibodies, and Alexa Fluor 488–conjugated anti–rabbit IgG antibody A-11034 (1:500, Molecular Probes), Cy3–conjugated anti–chick IgY antibody (1:500, Jackson ImmunoResearch), and Alexa Fluor 633–conjugated anti–mouse IgG antibody A-21052 (1:500, Molecular Probes) were used as secondary antibodies. Anti–Hts antibody 1B1 was used to visualize somatic cells, including cap cells and germline-specific organelles, or spectrosomes and fusomes. Anti–FasIII antibody was used to stain hub cells. Nuclei were visualized in embryos by staining with DAPI (2 μg/ml, Sigma-Aldrich) for 30 min.

Embryos, ovaries, and testes were mounted in either Vectashield (Vector Laboratories) or ProLong Diamond (Molecular Probes). Z-stack confocal images were taken on a Leica TCS-SP8 (Leica) confocal microscope. Optical slices were analyzed using the software, Fiji. Confocal serial images were analyzed to count the number of GFP-positive PGCs within each gonad (Fig. 1b) and the numbers of GFP-positive GSCs in each pair of ovaries/testes (Fig. 1d).

**Progeny production.** In experiments using y w hosts, each female or male adult derived from host embryos was mated with 5 y w males or 4–5 y w females, respectively, within one day after eclosion. The flies were subsequently transferred to new vials containing culture medium and allowed to lay eggs. Eggs obtained on days 1–3, 4–6, and 7–9 after mating were allowed to develop to adulthood, and the number of adult F1 progeny with red eyes was counted (Supplementary Fig. 2).

In the experiments using OvoA_OE hosts, we determined the fertility of host adults as follows. Each male host was mated with 4 y w females, and incubated in a vial with culture medium for 4 days (on days 1–4 after mating). On day 4, the male hosts producing first instar larvae were judged as fertile (Table 2). The single fertile male was mated with a single female host, and the female was subsequently allowed to lay eggs at 25 °C for 9 days (on days 1–9 after mating). Female hosts that produced F1 progeny were judged as fertile (Table 2). The phenotype of the F1 progeny produced from the fertile females on days 1–9 was observed under a dissecting microscope, and the number of F1 progeny counted. In a control experiment that excluded transplantation ("None" in Table 2), females and males were mated with y w flies to determine whether they were fertile or not.

**Statistics and reproducibility.** The statistical test used to analyze the data from each experiment is indicated in the figure legends. The experiments for all figures and tables were repeated more than three times, except twice for EG (10%) +

Sucrose (0.5 M) and DMSO (5%) + Sucrose (0.25 M) in Supplementary Table 1, and for *M17* 0d and *M17* 31–150d data in Tables 2, 3, and Fig. 2.

**Reporting summary**. Further information on research design is available in the Nature Research Reporting Summary linked to this article.

## Data availability

All data supporting the findings of this study are available within the article and its supplementary information files. Source data for Figs. 1b, d-f, and 2, Tables 1–3, and Supplementary Tables 1–2 is in Supplementary Data 1.

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

## Acknowledgements

We thank the KYOTO Stock Center and Bloomington *Drosophila* Stock Center for providing fly strains, and the Developmental Studies Hybridoma Bank for providing antibodies. This work was supported in part by a Grant-in-Aid for Scientific Research on Innovative Areas (JP18H05552) from the Japan Society for the Promotion of Science (JSPS) to S.K., a Grant-in-Aid for Scientific Research (C) (JP19K06780) from JSPS to T.T.-S. and M.T., a Grant-in-Aid for Challenging Exploratory Research (JP15K14441) from JSPS to D.T., grants (JP16km0210073, JP17km0210147, JP18km0210145) from Japan Agency for Medical Research and Development (AMED) to S.K., and by grants (JP16km0210072, JP17km0210146, JP18km0210146) to T.T.-S., and JP20km0210172 from AMED to T.T.-S. and S.K.

## Author contributions

Conceptualization: S.K.; Funding acquisition: M.T., T.T.-S., D.T. and S.K.; Methodology: M.A., Y.S., T.F., T.T.-S., D.T. and S.K.; Investigation: M.A., Y.S., T.F. and K.N.; Supervision: M.A., T.T.-S., D.T. and S.K.; Project administration: S.K; Writing - original draft: M.A.; Writing—review and editing: T.T.-S. and S.K.

## Competing interests

The authors declare no competing interests.
