## [Peer Review File · Communications Biology]

Reviewers' comments:

Reviewer #1 (Remarks to the Author):

The practical long-term genetic storage of *Drosophila* that is currently relying on living culture has been a challenging issue for many years. The authors established the cryopreservation protocol that keeps the transgenic fly lines by freezing their PGCs in preservative reagents, such as ethylene glycol and sucrose. The authors optimized the compositions of preservative reagents, evaluated the frozen conditions of PGCs, and quantified the efficiency of the method by counting the number of GSCs, PGCs and progenies. The authors also confirmed that the established method was able to be applied for the long-term (up to ~400 days) cryo-storage.

In general, the results were solid, and frozen PGC transplantation in CPA described in this study is a new technique for preserving fly strains, which be a great interest in the field. This method is useful to overcome the financial and spatial problems for keeping the fly living culture of the flies over the generations. However, I feel that some experiments and careful arguments are needed to prove advantages of this powerful technique. The biggest concern is about efficiency of the mutant PGCs. The method seems to be satisfied with the longer time preservation of *Drosophila* strains and potentially applied to any genetic strains. Indeed, there will be quite some demands to revive mutant stocks from their frozen PGCs which are collected from heterozygous and homozygous of a mutant allele, and occasionally, even homozygous of balancer chromosome. Therefore, the retrieval rate of mutant individuals would be lower than the cases reported in this study. Then how many donor embryos should be used to collect PGCs for successful reviving? Or how many PGCs should be preserved? In addition, after crossing into the balancers, expected genotype has to be selected by PCR or any other methods, if the mutant allele does not harbor any visible genetic markers. The authors should provide at least such argument for the practical use. I have also listed some concerns which need to be addressed to improve the manuscript.

1. Regarding to the transgenerational functions of the transplanted donor PGCs, not only F1 but also a couple of other generations should be examined for their fecundity, longevity or any such physiological activities.
2. Fig1 b and d were somehow swapped because the number of PGCs and GSCs are usually around 30 and 1-4, respectively.
3. I have an impression that the number of progenies of Fig2b is lower than expected; how many progenies will be obtained in the WT control?
4. Though the authors described gonads from the F-PGC were less able to colonize, the image data is missing. It should be provided with that of the control.

Reviewer #2 (Remarks to the Author):

Drosophila is unusual among widely-used animal model organisms in that no cryopreservation method presently exists, despite substantial past efforts to develop one. This means that in stock centres and individual laboratories, many thousands of fly strains need to be manually passaged onto new food every couple of weeks. This represents a huge volume of tedious work, with the consequent risks of strain loss, strain misidentification, and genetic drift.

This manuscript reports a new technology for cryopreservation of *Drosophila* strains based on freezing primordial germ cells. The evidence provided, while somewhat limited, makes a strong case for the feasibility and efficacy of this new technique. I would suggest that the authors consider the following, either in a revised version of this paper or in a subsequent follow-up study.

1. The strains used in the study are not as genetically compromised as many that are needed to be maintained long-term. It would be good to test the technique on some 'sick' strains such as double-balanced stocks, balanced stocks with large deficiencies, stocks with multiple mutations, etc. to see if it works as well on those. Certainly a more comprehensive pilot study will be necessary before this technique comes into widespread use.

2. The PI and his lab are among the world's leading experts in *Drosophila* pole cell transplantation, which is not really a widely-used technique. To assess the feasibility of bringing this method into widespread use, it would be good to include discussion of how long it would take someone with only fly husbandry and microinjection expertise to be trained to be efficient with this technique.

3. Related to the comment above, the manuscript states that the Kyoto Stock Centre is beginning to use this technique. I would like to see more discussion of how well this is going.

4. Finally, there are a few typographical errors. Line 271, should read Eggs not Egg. Line 279, should read four, not forth, and Fig., not Fi.

Reviewer #3 (Remarks to the Author):

In this manuscript, the author developed cryopreservation PGC method for backup living fly stocks. The author document that the transplanted PGCs are able to migrate into the host gonads, become GSCs, and produce functional gametes. In addition, different donor stains and long-term preservation do not reduce the above-described ability of transplanted PGCs. It is great that the author developed the method of PGC cryopreservation; however, there are some points that need to be addressed.

Major points:

1. Since this is a technique-based manuscript, the procedure and success rate of the method they developed should be described clearly in the text. For example, in table 1. The author injected F-PGCs into 125 host embryos. Is this number from a one-time injection? If it is only a one-time experiment, the author needs to confirm the result using at least three biological replicates. If the number is from multiple-time experiments, then the author should show the SD. Furthermore, how did the author suck out PGCs from embryos? What machine with what setting was used? how did the author control the number of PGCs (10-20) in the microinjection needle? How did they do transplantation? With what machine? And so on so forth.
2. Is there any reason to choose stage 5 embryos for collecting PGCs and for transplantation? How long does it take for stage 5 embryos to grow to stage 15?
3. If 20 PGCs was injected into each embryo, while each gonad only have ~3PGCs (fig. 1c), then the donor-PGCs successfully migrate to the gonad is only about 6/20 (30%). Similarly, the ratio of eclosion rate from host embryos is only about 30%. At the end, only ~9% of host embryos could become adults and produce donor-derived F1. This information should be mentioned or showed somewhere in the text/or table.
4. What is the ratio of progeny with red eyes per female?
5. How embryos were died due to the injection?
6. How did the author define embryo stage since the embryo has egg shield?
7. In the text, no description of donor-PGCs become male GSCs (fig. 1d). And why did the author only use female hosts, but not male hosts to get fig. 1e and f?
8. Sentence line 134-136 (page 6). How did the author which female carry donor-PGCs, since it requires ovary dissection and IHC.
9. Progeny derived from donor-PGCs should be tested for fertility and healthiness. It will be more convinced if the author can test their method with mutant or transgenic lines.
10. Success rate of cryopreservation of PGCs, embryos and ovaries should be compared and summarized in the text, so readers will appreciate the author's contribution more.

Minor points:

1. Hard to see the staining of Hts, FasIII and DAPI in Fig. 1c.
2. Sentence in line 117-118 (page 5) should add reference.
3. The sentence in line 121-123 (page 5) is very confusing. Has the Y chromosome of the donor been lost?

Our point-by-point response to the comments of the reviewers

1) Reviewer 2 stated that “*The strains used in the study are not as genetically compromised as many that are needed to be maintained long-term. It would be good to test the technique on some 'sick' strains to see if it works as well on those*” (**comment 1 by Reviewer 2**). These comments may be based on the concern of **Reviewer 1** that “*The biggest concern is about efficiency of the mutant PGCs. Indeed, there will be quite some demands to revive mutant stocks from their frozen PGCs which are collected from heterozygous and homozygous of a mutant allele, and occasionally, even homozygous of balancer chromosome. Therefore, the retrieval rate of mutant individuals would be lower than the cases reported in this study. In addition, after crossing into the balancers, expected genotype has to be selected by PCR or any other methods, if the mutant allele does not harbor any visible genetic markers. The authors should provide at least such argument for the practical use*” (**major comment by Reviewer 1**). Regarding the feasibility of our cryopreservation method, **Reviewer 1** further stated that “*I have an impression that the number of progenies of Fig2b is lower than expected; how many progenies will be obtained in the WT control?*” **Reviewer 3** stated that “*What is the ratio of progeny with red eyes per female?*”

Our response

We believe that these questions are all related to the feasibility of our method to cryopreserve fly strains. As suggested by the reviewers, when *y w* embryos are used as hosts to obtain progeny from the cryopreserved PGCs, it is necessary to distinguish donor-derived progeny from host-derived progeny using either dominant genetic markers or PCR. Furthermore, it is extremely difficult to reconstitute the original chromosome constitution from donor-derived progeny by mating. To overcome these problems, we decided to use an agametic *OvoA_OE* host. Using an agametic host enabled us to revive original donor strains from cryopreserved PGCs by inbreeding. This method reduces the labor required for selection and mating as mentioned by the reviewers. In addition, we used a weak (sick) strain (*M17*) as donor strain. *M17* is homozygous for two recessive mutations, heterozygous for three dominant mutations, and carries a balancer chromosome. We transplanted cryopreserved PGCs derived

from *M17* donor embryos into the agametic hosts to examine whether the original *M17* stock can be revived from the frozen PGCs. Our new data in **Tables 2, 3 and Fig. 2c, d** show that our new method using the agametic host is satisfactory for demands to revive unhealthy mutant stocks from cryopreserved PGCs. We added a statement concerning our new method and new data on **P7, L177 – P8, L206; P9, L209–218; P9, L230 – P10, L232 and L239–240; P13, L321–330**.

2) Reviewer 1 and Reviewer 3 stated as follows: “*Regarding to the transgenerational functions of the transplanted donor PGCs, not only F1 but also a couple of other generations should be examined for their fecundity, longevity or any such physiological activities*” (**comment 1 by Reviewer1**). “*Progeny derived from donor-PGCs should be tested for fertility and healthiness. It will be more convinced if the author can test their method with mutant or transgenic lines*” (**major point 9 by Reviewer 3**).

Our response

We agreed with these comments. We performed experiments to determine the fertility of F1 progeny derived from cryopreserved PGCs of *M17* donor strain. The results of these experiments are shown in **Fig. 2d and Table 3**. First, we found that the number of F2 progeny derived from cryopreserved PGCs of *M17* donor strain was similar to that obtained using flies from *M17* strain (**Fig. 2d**). Second, we found that the F1 flies were healthy, and could survive at least 9 d (**Fig. 2d legend**). Third, all F1 and F2 progeny derived from the cryopreserved *M17* PGCs were phenotypically identical to flies from the *M17* stock (**Table 3**). These data indicate that F1 progeny derived from donor PGCs are healthy and retain similar fecundity to that of the original donor stock. We added this statement on **P8, L192–196 and L200–204**.

3) Reviewer 1 stated that “*Fig1 b and d were somehow swapped because the number of PGCs and GSCs are usually around 30 and 1-4, respectively.*”

Our response

Figure 1b and 1d data are not swapped. The reviewer stated that the number of PGCs and GSCs are usually around 30 and 1–4, respectively. However, the correct number of PGCs per gonad is approximately 10, and the number of GSCs per adult is approximately 100 (female) or 16 (males).

4) Reviewer 1 stated that “*Though the authors described gonads from the F-PGC were less able to colonize, the image data is missing. It should be provided with that of the control.*”

Our response

We regret that this reviewer appears to have misunderstood our work. We do not think that the ability of F-PGCs to migrate into the gonads was reduced by freeze-thawing. We propose instead that almost all F-PGCs that were improperly freeze-thawed are eliminated in the host, whereas F-PGCs that are properly treated retain the ability to survive and migrate into the gonads, comparable to those of CPA-PGCs and naïve PGCs (P4, L102–107). For the sake of clarity, we changed the following sentences:

“These data show.... reduces the ability of PGCs to migrate into the embryonic gonads.”

↓

“These data show.... reduces the frequency of embryos with donor PGCs within the embryonic gonads.” (P4, L99–100)

“This is compatible.... F-PGCs were less able to colonize embryonic gonads than CPA-PGCs or Naïve-PGCs.”

↓

“This is compatible.... F-PGCs colonized embryonic gonads less frequently than CPA-PGCs or naïve PGCs.” (P5, L118–119)

5) **Reviewer 2** stated that *“To assess the feasibility of bringing this method into widespread use, it would be good to include discussion of how long it would take someone with only fly husbandry and microinjection expertise to be trained to be efficient with this technique.”*

Our response

As per the reviewer’s suggestion, we have added **Supplementary Table 5**, which summarizes the learning period for our cryopreservation method, to SI. This new table is cited on **P7, L174–176**.

6) **Reviewer 2** stated that *“Related to the comment above, the manuscript states that the Kyoto Stock Centre is beginning to use this technique. I would like to see more discussion of how well this is going.”*

Our response

Because the KYOTO Stock Center at KIT has just started to cryopreserve PGCs from some fly strains, we cannot calculate the number of fly strains cryopreserved per day. KYOTO Stock Center has been working hard to prepare data for this manuscript and for a video-based report.

7) **Reviewer 2** stated that *“There are a few typographical errors. Line 271, should read Eggs not Egg. Line 279, should read four, not forth, and Fig., not Fi.”*

Our response

We have corrected the typographical errors pointed out by the reviewer as follows:

P13, L318 (L271 in the original manuscript): Egg → Eggs

P14, L335 (L279 in the original manuscript): Fi → deleted

8) Reviewer 3 stated that “*Since this is a technique-based manuscript, the procedure and success rate of the method they developed should be described clearly in the text. For example, in table 1. The author injected F-PGCs into 125 host embryos. Is this number from a one-time injection? If it is only a one-time experiment, the author needs to confirm the result using at least three biological replicates. If the number is from multiple-time experiments, then the author should show the SD.*”

Our response

We performed our transplantation experiments more than four times each. The only exceptions were the experiments for Supplementary Table1 and *M17 0 d* and *M17 31–150 d* data (Tables 2, 3, and Fig. 2), which we performed only twice. We added a statement to clarify this under **Statistics and reproducibility** (P14, L334–336).

As per the reviewer’s suggestion, we have presented our data as mean \pm standard error (SEM) in **Supplementary Data 1**.

9) Reviewer 3 stated that “*how did the author control the number of PGCs (10-20) in the microinjection needle?*”

Our response

The number of PGCs transplanted into host embryos was counted under a microscope. We added this statement on **P11, L272–273**.

10) Reviewer 3 stated that “*Furthermore, how did the author suck out PGCs from embryos? What machine with what setting was used? how did the author control the number of PGCs (10-20) in the microinjection needle? How did they do transplantation? With what machine?*”

Our response

This information is extremely important for readers to perform cryopreservation in their own laboratories. We are currently preparing a video-based report on our detailed protocol. We have added a statement describing this on P9, L222–223.

11) Reviewer 3 stated that “*Is there any reason to choose stage 5 embryos for collecting PGCs and for transplantation?*”

Our response

We actually obtained PGCs from early stage 5 donor embryos (130–150 min AEL). This is the stage at which PGC formation is complete but somatic cellularization is not. We could therefore obtain PGCs while avoiding contamination by somatic cells. We added this statement on P10, L254 – P11, L257.

12) Reviewer 3 stated that “*How long does it take for stage 5 embryos to grow to stage 15?*”

Our response

Stage 5 embryos take approximately 20 h to reach stage 15 at 18°C. Thus, the transplanted embryos were allowed to develop at 18°C for 20 h. We added this statement on P12, L287–288.

13) Reviewer 3 stated that “*If 20 PGCs was injected into each embryo, while each gonad only have ~3PGCs (fig. 1c), then the donor-PGCs successfully migrate to the gonad is only about 6/20 (30%). Similarly, the ratio of eclosion rate from host embryos is only about 30%. At the end, only ~9% of host embryos could become adults and produce donor-derived F1. This information should be mentioned or showed somewhere in the text/or table.*”

Our response

In our protocol using *yw* hosts, the efficiency of PGC transplantation to obtain single adult host producing donor-derived F1 progeny was 8.8% [11/125 (F-PGC)], 19.4% [13/67 (CPA-PGC)], and 13.5% [19/141 (naïve PGC)]. This statement was added in **Table 1 footnote** (P24, L531–533). Furthermore, in our protocol using agametic *OvoA_{OE}* hosts, transplantation efficiency (no. of host couples producing F1 progeny / no. of transplanted embryos) was 12.1%, when transplanted with F-PGCs from *MI7* embryos. We have added a statement to the footnote for **Table 2** (P25, L547–551).

14) Reviewer 3 asked, “*How embryos were died due to the injection?*”

Our response

A subset of embryos usually dies after either PGC transplantation or micromanipulation. Presumably, they are eliminated due to desiccation. The survival rate of embryos until stage 15 was 35.0% (F-PGCs), 47.4% (CPA-PGCs), and 50.0% (naïve PGCs). These survival rates are not significantly different among F-PGC, CPA-PGC, and naïve PGC, suggesting that the subset of embryos died regardless of whether the embryos are freeze-thawed or CPA treated. We added this statement to the footnote for **Supplementary Table 2** (SI, P7, L86–88).

15) Reviewer 3 asked, “*How did the author define embryo stage since the embryo has egg shield?*”

Our response

Embryos were dechorionated, aligned on double-sided adhesive tape, and covered with a drop of silicone oil for PGC transplantation. These treatments enabled us to observe internal structures of embryos under a dissection microscope. Thus, we defined the embryonic stages by observing embryonic morphologies according to

Compos-Ortega and Hartenstein (1985). We added these statements to “Freeze-thawing PGCs” (P10, L251–252) and “Immunostaining” (P12, L287–289).

16) Reviewer 3 stated that “*In the text, no description of donor-PGCs become male GSCs (fig. 1d). And why did the author only use female hosts, but not male hosts to get fig. 1e and f?*”

Our response

We added the sentence “The number of GSCs derived from F-PGCs was similar to GSCs originating from CPA-PGCs or naive PGCs in both females and males (Fig. 1d)” on P6, L138.

The reason why we only used females to obtain data for Fig. 1e and f was that, once male flies mate with females, males are rejected by post-coital females, and the males court less avidly when paired with virgin females (courtship memory).

17) Reviewer 3 stated that “*Sentence line 134-136 (page 6). How did the author which female carry donor-PGCs, since it requires ovary dissection and IHC?*”

Our response

After mating, the adult female hosts were allowed to lay eggs for 9 d to determine donor-derived offspring production. On day 10, ovaries were dissected from adult hosts to count the donor-derived GSCs. To help describe our experimental procedure, we added Supplementary Fig. 2.

18) Reviewer 3 stated that “*Success rate of cryopreservation of PGCs, embryos and ovaries should be compared and summarized in the text, so readers will appreciate the author’s contribution more.*”

Our response

We could not compare the success rates of our method with previous cryopreservation methods using embryos and ovaries because the previous methods

suffered from poor reproducibility. Hence, no one has succeeded in cryopreserving fly strains thus far using these previous methods.

19) Reviewer 3 stated that “*Hard to see the staining of Hts, FasIII and DAPI in Fig. 1c.*”

Our response

In the revised manuscript (**Fig. 1c**), the separated, but not merged, image of each signal for DAPI, Vasa, and Hts/FasIII was shown. Hts and FasIII signals could not be distinguished from each other because both the anti-Hts antibody and the anti-FasIII antibody were detected with the same secondary antibody.

20) Reviewer 3 stated that “*Sentence in line 117-118 (page 5) should add reference.*”

Our response

As suggested by the reviewer, we added a reference to the sentence on P5, L117–118 in the original manuscript (**P5, L121–122** in the revised manuscript).

21) Reviewer 3 stated that “*The sentence in line 121-123 (page 5) is very confusing. Has the Y chromosome of the donor been lost?*”

Our response

To avoid misunderstanding, we changed the sentence (**P5, L126–128**) to the following:

“This ability is particularly important considering that male-specific Y-chromosomes of donors can be recovered only from male F1 progeny derived from male F-PGCs.”

↓

“This ability is particularly important considering that male-specific Y-chromosomes of donors can be obtained only from male F1 progeny derived from male F-PGCs.”

22) In accordance with the Author's guidelines, we have made the following changes:

In Fig. 1b, d, and f, bar graphs were converted to "beeswarm" figures.

In Fig. 2b, Median values were added.

In Fig. 2 legend, Table 1 footnote, and Supplementary Table 2 footnote

"Fisher's exact test"

↓

"two-sided Fisher's exact test"

REVIEWERS' COMMENTS:

Reviewer #1 (Remarks to the Author):

General comments

The manuscript has been significantly improved by overcoming the difficulty of segregating donor strains from the host agametic strains. However, compared to the recently-reported cryopreservation method of embryos (ref. 24), the PGC-based cryopreservation seems to be less versatile; the transplantation efficiency and its transplantations process and is relevant to the well-trained person who is doing transplantation PGCs. The authors mentioned making a video-based report shortly, which would be very helpful to learn the manipulation of the frozen PGCs preparation and transplantation them to the embryos. I strongly suggest the authors to consider having a clearer statement or an argument for any advantages of using the PGC-based cryopreservation method compared with the recently published cryopreservation method of embryos, in order to encourage the fly community to use PGC-based cryopreservation.

Point-to point responses to Authors comments:

1) Reviewer 2 stated that "The strains used in the study are not as genetically compromised as many that are needed to be maintained long-term. It would be good to test the technique on some 'sick' strains to see if it works as well on those" (comment 1 by Reviewer 2). These comments may be based on the concern of Reviewer 1 that "The biggest concern is about efficiency of the mutant PGCs. Indeed, there will be quite some demands to revive mutant stocks from their frozen PGCs which are collected from heterozygous and homozygous of a mutant allele, and occasionally, even homozygous of balancer chromosome. Therefore, the retrieval rate of mutant individuals would be lower than the cases reported in this study. In addition, after crossing into the balancers, expected genotype has to be selected by PCR or any other methods, if the mutant allele does not harbor any visible genetic markers. The authors should provide at least such argument for the practical use" (major comment by Reviewer 1). Regarding the feasibility of our cryopreservation method, Reviewer 1 further stated that "I have an impression that the number of progenies of Fig2b is lower than expected; how many progenies will be obtained in the WT control?" Reviewer 3 stated that "What is the ratio of progeny with red eyes per female?"

Our response

We believe that these questions are all related to the feasibility of our method to cryopreserve fly strains. As suggested by the reviewers, when y w embryos are used as hosts to obtain progeny from the cryopreserved PGCs, it is necessary to distinguish donor-derived progeny from host-derived progeny using either dominant genetic markers or PCR. Furthermore, it is extremely difficult to reconstitute the original chromosome constitution from donor-derived progeny by mating. To overcome these problems, we decided to use an agametic OvoA_OE host. Using an agametic host enabled us to revive original donor strains from cryopreserved PGCs by inbreeding. This method reduces the labor required for selection and mating as mentioned by the reviewers. In addition, we used a weak (sick) strain (M17) as donor strain. M17 is homozygous for two recessive mutations, heterozygous for three dominant mutations, and carries a balancer chromosome. We transplanted cryopreserved PGCs derived from M17 donor embryos into the agametic hosts to examine whether the original M17 stock can be revived from the frozen PGCs. Our new data in Tables 2, 3 and Fig. 2c, d show that our new method using the agametic host is satisfactory for demands to revive unhealthy mutant stocks from cryopreserved PGCs. We added a statement concerning our new method and new data on P7, L177 – P8, L206; P9, L209–218; P9, L230 – P10, L232 and L239–240; P13, L321–330.

The authors used an agametic host, OvoA_OE, which enables to revive of original donor strains from cryopreserved PGCs by inbreeding and reduces the labor required for selection and mating. OvoA_OE host is quite useful because the retrieving rate is comparable to that of yw. This is a significant improvement of the PGC-based cryopreserve method for isolation of only donor strains after cryopreservation and transplantation of PGCs into host embryos without any costs of other laborious selection methods.

Related comments

1. The point that readers would like to know is the PGC transplantation efficiency that produces

donor-derived F1. The transplantation efficiency of each donor (8.8%, 19.4%, 13.5% in table 1 /12.1%, 9.7%, 6.8% in table2) should be in the table. Ideally, evaluation items may better be in the same format to avoid confusion.

2. I wonder freeze-thawed embryos can be robust for the repeated cryopreservation cycles and would like to know the possibility to test this.

2) Reviewer 1 and Reviewer 3 stated as follows: "Regarding to the transgenerational functions of the transplanted donor PGCs, not only F1 but also a couple of other generations should be examined for their fecundity, longevity or any such physiological activities" (comment 1 by Reviewer1). "Progeny derived from donor-PGCs should be tested for fertility and healthiness. It will be more convinced if the author can test their method with mutant or transgenic lines" (major point 9 by Reviewer 3).

Our response

We agreed with these comments. We performed experiments to determine the fertility of F1 progeny derived from cryopreserved PGCs of M17 donor strain. The results of these experiments are shown in Fig. 2d and Table 3. First, we found that the number of F2 progeny derived from cryopreserved PGCs of M17 donor strain was similar to that obtained using flies from M17 strain (Fig. 2d). Second, we found that the F1 flies were healthy, and could survive at least 9 d (Fig. 2d legend). Third, all F1 and F2 progeny derived from the cryopreserved M17 PGCs were phenotypically identical to flies from the M17 stock (Table 3). These data indicate that F1 progeny derived from donor PGCs are healthy and retain similar fecundity to that of the original donor stock. We added this statement on P8, L192–196 and L200–204.

The additional items, fecundity, healthiness, and phenotypic identity of cryopreserved PGCs of M17 donor strain, are satisfactory.

3) Reviewer 1 stated that "Fig1 b and d were somehow swapped because the number of PGCs and GSCs are usually around 30 and 1-4, respectively."

Our response

Figure 1b and 1d data are not swapped. The reviewer stated that the number of PGCs and GSCs are usually around 30 and 1–4, respectively. However, the correct number of PGCs per gonad is approximately 10, and the number of GSCs per adult is approximately 100 (female) or 16 (males).

Now I understood that the authors mentioned GSCs per adult, not per ovariole in Figure1.

4) Reviewer 1 stated that "Though the authors described gonads from the F-PGC were less able to colonize, the image data is missing. It should be provided with that of the control."

Our response

We regret that this reviewer appears to have misunderstood our work. We do not think that the ability of F-PGCs to migrate into the gonads was reduced by freeze-thawing. We propose instead that almost all F-PGCs that were improperly freeze-thawed are eliminated in the host, whereas F-PGCs that are properly treated retain the ability to survive and migrate into the gonads, comparable to those of CPA-PGCs and naïve PGCs (P4, L102–107). For the sake of clarity, we changed the following sentences:

"These data show.... reduces the ability of PGCs to migrate into the embryonic gonads."

↓

"These data show.... reduces the frequency of embryos with donor PGCs within the embryonic gonads." (P4, L99–100)

"This is compatible.... F-PGCs were less able to colonize embryonic gonads than CPA-PGCs or Naïve-PGCs."

↓

"This is compatible.... F-PGCs colonized embryonic gonads less frequently than CPA-PGCs or naïve PGCs." (P5, L118–119)

I understood the descriptions and those link to the (P4, L99–100) and (P5, L118–119). “We propose instead that almost all F-PGCs that were improperly freeze- thawed are eliminated in the host, whereas F-PGCs that are properly treated retain the ability to survive and migrate into the gonads, comparable to those of CPA-PGCs and naïve PGCs.” The corrections are also now clear.

In addition, though the authors did not provide the comments, I found the answers to my questions below in the manuscripts.

Then how many donor embryos should be used to collect PGCs for successful reviving? Or how many PGCs should be preserved?^

“transplanted host embryos were required to obtain single host couple producing donor-derived progeny of both sexes [no. of embryos transplanted / no. of couples producing donor-derived F1 progeny: 74/5 ~ 15 (M17 31–150d)]. Given that 10–20 PGCs were transplanted into a host embryo (seeMethod), 150–300 PGCs are required to produce a single fertile couple, and this number of PGCs can be collected into one glass needle from 27 donor embryos. This PGC collection and subsequent PGC transplantation into agametic host embryos can be performed within 50 and 100 minutes (min), respectively.”

Reviewer #2 (Remarks to the Author):

My concerns have been fully addressed in this revised version.

Reviewer #3 (Remarks to the Author):

The author has fully addressed my questions. The article is suitable for publishing in Communication Biology.

Our point-by-point response to the comments of the reviewers

1) Reviewer 1 stated that *“I strongly suggest the authors to consider having a clearer statement or an argument for any advantages of using the PGC-based cryopreservation method compared with the recently published cryopreservation method of embryos, in order to encourage the fly community to use PGC-based cryopreservation.”* (**General comment by Reviewer 1**). Reviewer 1 further stated that *“I wonder freeze-thawed embryos can be robust for the repeated cryopreservation cycles and would like to know the possibility to test this”*(**Related comment 2 by Reviewer 1**).

Our response

The recently published report (Zhan et al., 2021) have revealed that embryo-based cryopreservation method requires the embryos in a very narrow range of ages optimal for cryopreservation. In contrast, our method uses PGCs obtained from the blastodermal embryos, which can be easily collected. Regarding this point, PGC-based cryopreservation method might be more advantageous than the other method. Anyway, we will compare the efficiency and feasibility between these two cryopreservation methods, in the future. These statement was added in the last part of our manuscript (P9, L225-P10, L234).

2) Reviewer 1 stated that *“The point that readers would like to know is the PGC transplantation efficiency that produces donor-derived F1. The transplantation efficiency of each donor (8.8%, 19.4%, 13.5% in table 1 /12.1%, 9.7%, 6.8% in table2) should be in the table. Ideally, evaluation items may better be in the same format to avoid confusion.”* (**Related comment 1 by Reviewer 1**).

Our response

We agreed with this comment. We therefore put "PGC transplantation efficiency" in **Table 1** and **2**.